Resting heart rate and risk of dementia: a Mendelian randomization study in the international genomics of Alzheimer’s Project and UK Biobank

Chen Xingxing 1 2
Zheng Yi 3
http://orcid.org/0000-0002-4407-4445 Wang Jun 4
http://orcid.org/0000-0002-6009-9714 Yue Blake 5 6
http://orcid.org/0009-0009-7803-5581 Zhang Xian 2
Nakai Kenta 3 7 knakai@ims.u-tokyo.ac.jp
Yan Lijing L. 1 2 8 9 lijing.yan@dukekunshan.edu.cn
1 School of Public Health, Wuhan University , Wuhan, Hubei Province , China
2 Duke Kunshan University, Global Health Research Center , Kunshan, Suzhou , China
3 The University of Tokyo, Department of Computational Biology and Medical Science , Kashiwa , Japan
4 Huazhong University of Science and Technology, Department of Otorhinolaryngology of Union Hospital , Wuhan, Hubei Province , China
5 School of Business and Law, Edith Cowan University , Perth, WA , Australia
6 National Institute for Stroke and Applied Neurosciences, Auckland University of Technology , Auckland , New Zealand
7 The University of Tokyo, The Institute of Medical Science , Tokyo , Japan
8 Duke University, Duke Global Health Institute , Durham, North Carolina , United States of America
9 Peking University, Institute for Global Health and Management , Beijing , China
Liu Feng
Electronic publication date: 2024 Mar 15
Publication date: 2024
Volume: 12
Electronic Location ID: e17073
Received 2023 Sep 6; Accepted 2024 Feb 18
Copyright: © 2024 Chen et al.
Copyright year: 2024
Copyright holder: Chen et al.
License: This is an open access article distributed under the terms of the Creative Commons Attribution License, which permits unrestricted use, distribution, reproduction and adaptation in any medium and for any purpose provided that it is properly attributed. For attribution, the original author(s), title, publication source (PeerJ) and either DOI or URL of the article must be cited.
License URL: https://creativecommons.org/licenses/by/4.0/

Keywords: Dementia, Alzheimer’s disease, Resting heart rate, Genetic variants, Mendelian randomization

Funding: There is no funding role in this study.

==============================
Background

Observational studies have demonstrated that a higher resting heart rate (RHR) is associated with an increased risk of dementia. However, it is not clear whether the association is causal. This study aimed to determine the causal effects of higher genetically predicted RHR on the risk of dementia.

Methods

We performed a two-sample Mendelian randomization analysis to investigate the causal effect of higher genetically predicted RHR on Alzheimer’s disease (AD) using summary statistics from genome-wide association studies. The generalized summary Mendelian randomization (GSMR) analysis was used to analyze the corresponding effects of RHR on following different outcomes: 1) diagnosis of AD (International Genomics of Alzheimer’s Project), 2) family history (maternal and paternal) of AD from UK Biobank, 3) combined meta-analysis including these three GWAS results. Further analyses were conducted to determine the possibility of reverse causal association by adjusting for RHR modifying medication.

Results

The results of GSMR showed no significant causal effect of higher genetically predicted RHR on the risk of AD (βGSMR = 0.12, P = 0.30). GSMR applied to the maternal family history of AD (βGSMR = −0.18, P = 0.13) and to the paternal family history of AD (βGSMR = −0.14, P = 0.39) showed the same results. Furthermore, the results were robust after adjusting for RHR modifying drugs (βGSMR = −0.03, P = 0.72).

Conclusion

Our study did not find any evidence that supports a causal effect of RHR on dementia. Previous observational associations between RHR and dementia are likely attributed to the correlation between RHR and other cardiovascular diseases.

Introduction

Dementia is one of the rising public health challenges affecting 43.8 million people in 2016 worldwide (Nichols et al., 2019, pp. 1990–2016). The number of individuals with dementia is estimated to be 152.8 million by 2050, with nearly two-thirds of them residing countries with low- and middle-incomes (Nichols et al., 2022). Dementia severely affects patients’ life qualities, families, and health-care systems as a whole (Etters, Goodall & Harrison, 2008). Without cures, identification of the modifiable risk factors is important for postponing the development of dementia.

Despite the mounting global effect of dementia, the cause remains largely unknown (Wirdefeldt et al., 2011; O’Gorman, Lucas & Taylor, 2012; Baumgart et al., 2015). Previous evidence has confirmed that cardiovascular diseases (CVDs) along with their modifiable risk factors including diabetes, obesity, smoking, and hypertension, are associated with dementia (Chen et al., 2021; Korologou-Linden et al., 2022). Still, these factors may only recapitulate partial aspects of dementia’s development. Recently, there has been growing recognition of the predictivity of resting heart rate (RHR) beyond CVD, particularly for dementia (Bohm et al., 2012). Two recent prospective studies have shown that elevated RHR is associated with increased risks of cognitive decline and dementia in long-term follow-up (Imahori et al., 2022; Deng et al., 2022). RHR is simple and easily obtained, so it should prove useful to help identify potentially high-risk populations of dementia in a wide variety of settings. RHR is also manageable via exercise and medications. Therefore, a reduction in RHR through exercise or medical treatment may be explored as an intervention target to delay cognitive decline, which might have crucial implications in public health. However, their relationships have courted controversy as null associations are yielded (Swedberg et al., 2010; Haring et al., 2020). Besides, the causal effect of higher RHR on cognitive decline and dementia remains to be investigated. This is important as a higher RHR may be linked to dementia via an indirect pathway of CVDs in observational study, e.g., ischemic heart disease and stroke (Kuźma et al., 2018; Kokkinidis et al., 2020).

These controversy in current literature necessitate the adoption of alternative models, particularly those that emphasize the causal associations and make use of multiple data instead of a single cohort. Mendelian randomization (MR) is a method that can be used to estimate the causal association of the modifiable exposure on the outcome. It utilizes genetic variants to establish a robust link. It also takes the advantage of large summary statistics derived from genome-wide association study (GWAS) (Hemani, Bowden & Davey Smith, 2018).

In the present study, two-sample MR analyses were performed to determine if there is a causal effect of higher genetically predicted RHR on dementia. Data were obtained from a set of multiple sources, including large-scale GWAS analyses on RHR (Eppinga et al., 2016), Alzheimer’s disease (AD) from the International Genomics of Alzheimer’s Project (IGAP) (Lambert et al., 2013), and the UK Biobank (UKBB) (Marioni et al., 2018).

Materials and Methods

Data sources for RHR and AD

This study was conducted based on de-identified summary statistics that are publicly available (the hitherto largest GWAS of RHR, the International Genomics of Alzheimer’s Project (IGAP) GWAS), and the GWAS of family history from UKBB. The present analyses were conducted using genetic variants which were associated with RHR from a previously published GWAS of RHR in which a total of 64 single-nucleotide polymorphisms (SNPs) were identified from 265,046 individuals, explaining 2.5% of the variance in RHR (Eppinga et al., 2016). A total of three GWAS summary statistics were used to determine the link between RHR genetic variation and AD.

The IGAP AD GWAS dataset was selected to conduct the primary analysis (Lambert et al., 2013). Diagnosed AD was used by IGAP GWAS as an outcome, and all patients with AD satisfied the NINCDS-ADRDA criteria or DSM-IV guidelines.

We additionally used the GWAS dataset from UKBB family history of AD, including maternal family history (MA-U) and paternal family history (FA-U), to conduct the MR analysis (Marioni et al., 2018). In this dataset, a proxy phenotype of self-reported family history of AD/dementia was applied to represent the AD case-control status. The accuracy and reliability of such self-reported measurements, confirmed through genetic correlation analysis in a global meta-analysis, indicated that approximately 81% of the genomic inflation (λ = 1.11) was attributed to the polygenic signal rather than confounding from self-report (Marioni et al., 2018). Finally, the GWAS dataset from a meta-analysis with the combination of the IGAP AD GWAS and family history of AD of UKBB (FH-AD) was selected to further confirm the primary results. Among all individuals included in the RHR GWAS study, 89.1% are British White, 2.7% are Irish White, and the remaining individuals have different ethnic backgrounds. For the outcomes used in the sensitivity analysis, there may be an overlap range of 0 to 40% in samples (Eppinga et al., 2016). However, the data for IGAP and UKBB are generated from two distinct sampled populations. Therefore, there is no sample overlap in the primary analysis, and we consider the findings of this study to be reliable. Full GWAS and variable descriptions were shown in the Supplemental Materials.

Criteria for instrument selection

Genetic instruments were selected based on the following criteria: (1) a collection of SNPs related with RHR reached GWAS significance (P < 5 × 10−8) (Hemani, Bowden & Davey Smith, 2018); and (2) all selected SNPs were previously confirmed in prior studies (Eppinga et al., 2016). According to the assumptions of MR analysis (Davey Smith & Ebrahim, 2004), in order to examine the previously observed associations between genetic instruments and AD and remove potential confounders, a search on the PhenoScanner GWAS platform (https://github.com/phenoscanner/phenoscanner; version 2) was performed (Staley et al., 2016) and SNPs that have been reported to be significantly (P < 5 × 10−8) associated with dementia as well as a SNP with high blood pressure in published GWASs were removed before analyses. For harmonizing data, we verified the effect/reference allele of all selected SNPs and ensure their consistency with the dbSNP151 reference. All non-matching alleles were exchanged and the sign of the beta estimates was altered. All reciprocal strand alleles were screened for quality control, and excluded ambiguous variants (i.e., alleles either A/T or C/G) and SNPs with allelomorph frequencies from 0.4 to 0.6. Detailed information of all genetic instruments and the description of the criteria for SNP exclusion process were shown in the Supplemental Materials.

Mendelian randomization analysis

For the principal test of our hypothesis, the GSMR R package was used to perform a generalized summary MR (GSMR) (Zhu et al., 2018). To further select instrumental variables based on linkage disequilibrium, we used the gcta tool to construct an LD correlation matrix with the data from the European population in the 1000 Genomes Project (1000 Genomes Project Consortium, 2010) as the ld_rho parameter in GSMR. We also set the ld_r² and ld_fdr thresholds at 0.05. SNPs that show pleiotropy by the HEIDI-outlier instruments analysis (P < 0.01) were removed. To examine the robustness of the results and to exclude the potential influence of RHR modifying medication on the interaction between RHR and AD, we conducted an analysis excluding individuals using RHR modifying medication (beta-blockers or calcium-channel blockers (N = 11,405)) (Eppinga et al., 2016).

Subsequently, we conducted several sensitivity analyses for exposure on outcome effects was performed using the TwoSampleMR R package (Hemani et al., 2018). The fixed effect inverse variance–weighted (IVW) meta-analysis, random effect IVW and MR-Egger regression approach were employed for MR analysis. The IVW is identical to a weighted regression of exposure on an outcome with the interception restricted to zero. The MR-Egger is a MR method that can be employed when the instrumental variable assumptions do not hold. The slope coefficient from the regression provides an estimate of the causal effect. If the confounders work in the causal pathways on the outcome beyond the exposure, the analysis could provide biased results with horizontal pleiotropy for instrument variants (Bowden, Davey Smith & Burgess, 2015). Therefore, we compared the results from IVW and MR-Egger to test whether the MR findings were influenced by pleiotropic effects. We also conducted a bidirectional MR analysis to mitigate potential reverse causation.

Although MR-Egger method is relatively robust when estimating the effect with the presence of pleiotropy, it can also be affected by a weak statistical power (Hemani, Bowden & Davey Smith, 2018). However, MR-Egger regression estimate was also known to be the first choice when pleiotropy was detected. The IVW-MR method was preferred when there is no pleiotropy. Furthermore, we performed the modified Cochran’s Q test and the leave-one-SNP-out analyses to detect heterogeneous outcomes using TwoSampleMR R package (Hemani et al., 2018).

We also calculated the post hoc power after all the analyses for each MR results were completed (Burgess, 2014). The coefficient (R2) of exposure on genetic variants was determined on the total allele score using the TwoSampleMR R package (Hemani et al., 2018). Additionally, we assessed the instrumental strength of the forty-three RHR genetic variants by computing the first-stage F-statistic. The F-statistic was utilized to estimate the minimum association magnitudes detectable in our analysis. We employed a web-based tool (https://sb452.shinyapps.io/power/) to access the statistical power of GSMR results, adhering to a two-sided type-I error rate of α = 0.05 (Burgess, 2014).

Results

A total of 43 SNPs reached the GWAS significance (P < 5 × 10−8) and were extracted from the RHR GWAS summary data (Eppinga et al., 2016). Detailed genetic instrument filtering process was shown in Table S1. Several sensitivity analyses were conducted to determine the effect of using different types of MR methods, IVW-MR and MR-Egger, as well as the influence of pleiotropy.

A total of 35 SNPs met the criteria for instrument selection and were used for analysis of the IGAP GWAS dataset. Results using IGAP dataset showed no significant causal effect of higher genetically predicted RHR on the risk of AD (βGSMR = 0.12, P = 0.30) (Fig. 1A).

Figure 1 Generalized summary Mendelian randomization (GSMR) analysis of association of resting heart rate (RHR) with Alzheimer’s disease (AD) diagnosis and family history of AD.

(A) IGAP AD diagnosis with resting heart rate. (B) UKBB maternal family history with resting heart rate. (C) UKBB paternal family history with resting heart rate. (D) UKBB and IGAP meta-analysis with resting heart rate.

There was no indication of pleiotropic SNP from MR-PRESSO for all outcome analyses (P = 0.96 for IGAP, P = 0.89 for MA-U, P = 0.85 for FA-U, P = 0.93 for FH-AD by Marioni et al. (2018), and P = 0.81 for FH-AD by Kunkle et al., 2019). No confounding heterogeneity was detected in all analyses (leave-one-SNP-out; Cochran’s Q statistic, P = 0.96) in this case (Fig. S1). Additional analysis was performed by excluding the SNPs with ambiguous strand which had been analyzed in previous analyses. The results from the alternative analysis were not significantly different from the previous results in the IGAP. There is no evidence of a source of pleiotropy through conducting sensitivity analysis (MR-Egger intercept, P = 0.93). Both the MR-Egger and IVW-MR with fixed or random effect verified the results of GSMR approach for RHR ( β^xy-IVWfixed = 0.12, P = 0.30; β^xy-IVWrandom = 0.12, P = 0.18; β^xy-MR-egger = 0.19, P = 0.25) (Fig. S2).

Despite of a protective effect in the UKBB family history GWAS, estimates confirmed that there was no significant causal relationship between RHR and the family history of AD: MA-U (βGSMR = −0.18, P = 0.13), FA-U (βGSMR = −0.14, P = 0.39) (Figs. 1B–1C). The results from sensitivity analyses (Cochran’s Q statistic, PFA-U = 0.72) and pleiotropy analysis (MR-Egger intercept, PMA-U = 0.83) determined no presence of confounding heterogeneity of effect sizes. In addition, the results of IVW-MR were consistent with the GSMR analysis in this case (MA-U ( β^xy-IVWfixed = −0.18, P = 0.13), FA-U ( β^xy-IVWfixed = −0.14, P = 0.39), MA-U ( β^xy-IVWrandom = −0.18, P = 0.08), FA-U ( β^xy-IVWrandom = −0.14, P = 0.33) (Table 1).

Table 1 Summary results of the MR analysis for RHR on dementia.

Methods	RHR- IGAP	RHR–MA-U	RHR–FA-U	RHR-FH-AD	
	Number of SNPs	β (SE)	P	Number of SNPs	β (SE)	P	Number of SNPs	β (SE)	P	Number of SNPs	β (SE)	P	
GSMR	35	0.12 (0.12)	0.30	38	−0.18 (0.12)	0.13	37	−0.14 (0.17)	0.39	36	−0.03 (0.08)	0.72	
IVW (Fixed effects)	35	0.12 (0.11)	0.30	38	−0.18 (0.12)	0.13	37	−0.14 (0.17)	0.39	36	−0.03 (0.07)	0.71	
IVW (Random effects)	35	0.12 (0.09)	0.18	38	−0.18 (0.10)	0.08	37	−0.14 (0.14)	0.33	36	−0.03 (0.06)	0.66	
MR-Egger	35	0.19 (0.16)	0.25	38	−0.45 (0.33)	0.18	37	−0.28 (0.24)	0.25	36	−0.16 (0.21)	0.42	
Notes:

The same SNPs were used in each outcome dataset to compare the results between the GSMR analysis and the other MR analysis. In all analyses, there was no horizontal pleiotropy (All MR-Egger Ps > 0.05). The IVW method was additionally used to determine the causal effect. No heterogeneity was observed as Cochran’s Q statistics P value > 0.10 in all analyses.

MR, Mendelian randomization; AD, Alzheimer’s disease; RHR, resting heart rate; GSMR, generalized summary Mendelian randomization; IGAP, International Genomics of Alzheimer’s Project; IVW, inverse variance weighted; FH-AD, a GWAS dataset from a combined meta-analysis; MA, maternal family history; FA, paternal family history; SNP, single nucleotide polymorphism.

Subsequently, we conducted an overall analysis of RHR on the meta-analysis with combination of IGAP and UKBB family history (FH-AD) (Marioni et al., 2018). A total of 36 SNPs overlapped between RHR and the FH-AD dataset. We confirmed that there was no significant causal effect of higher RHR on AD (βGSMR = −0.03, P = 0.72; β^xy-IVWfixed = −0.03, P = 0.72, β^xy-IVWrandom = −0.03, P = 0.66) (Fig. 1D; Table 1). The pleiotropy and heterogeneity were not detected in this case. (MR-Egger intercept, P = 0.47; Cochran’s Q statistic, P = 0.91). The reverse MR analysis confirmed that there is no significant causal effect of AD on RHR (Fig. S4). Overall, the results of bidirectional MR confirmed the robustness of the primary results (Table S4).

Further analyses adjusting RHR-modifying medication were estimated in the IGAP and UKBB family history datasets. The forest plot shows both a generalized summary of Mendelian randomization (GSMR) and the outcome of the MR-Egger estimates of the relationship between RHR and dementia by adjusting the use of RHR inhibitors (Fig. 2). Participants who were taking RHR-inhibitor medications were excluded to avoid reverse causal effect in these analyses. The results showed that RHR was not associated with AD diagnosis and family history of AD from UKBB (Fig. 2; Supplemental Materials).

Figure 2 Forest plot of Mendelian randomization (MR) estimate of the relationship between resting heart rate (RHR) and Alzheimer’s disease diagnosis (AD, International Genomics of Alzheimer’s Project, (IGAP)) or family history of AD from UK Biobank (UKBB).

The selected instrumental variables accounted for approximately 0.32% to 0.35% of the phenotypic variance (Table 2). The corresponding F-statistics for the datasets from IGAP, MA-U, FA-U, FH-AD by Marioni, and FH-AD by Kunkle et al. (2019) were 850.45, 895.12, 904.63, 870.45, and 920.41, respectively (Table 2). The calculated statistical powers were 10.2%, 40.1%, 16.7%, 6.3%, and 12.8%, respectively (Table 2). These results indicate that, within the sample size utilized in this analysis, the detection of a causal effect of high resting heart rate on the incidence of Alzheimer’s disease was not effectively achievable.

Table 2 Power (two-sided α = 0.05) for two-sample mendelian randomization analysis.

Exposure	Outcome-AD
(Dataset)	Participants in outcome	Proportion of cases in the outcome dataset	Causal Effect
(exp(βxy))	R2 of
instrument	F statistics	Power
(observed effect)	
RHR	IGAP	54,162	0.31	1.12	0.32%	850.45	10.2%	
RHR	MA-U	314,278	0.10	0.84	0.34%	895.12	40.1%	
RHR	FA-U	314,278	0.05	0.87	0.34%	904.63	16.7%	
RHR	FH-AD †	388,324	0.19	0.97	0.33%	870.45	6.3%	
RHR	FH-AD §	63,926	0.34	0.89	0.35%	920.41	12.8%	
Notes:

Post-hoc power calculations were based on the method developed by Burgess (2014).8 Causal Effect = exponentiated estimate obtained from GSMR.

† (FH-AD) GWAS conducted by Marioni et al. (2018).

§ (FH-AD 2) GWAS conducted by Kunkle et al. (2019).

AD, Alzheimer’s disease; RHR, resting heart rate; GWAS, genome-wide association study; IGAP, International Genomics of Alzheimer’s Project; FH-AD, a GWAS dataset from a combined meta-analysis; MA, maternal family history; FA, paternal family history.

Discussion

In the present study, a two-sample MR analysis was performed using previously published GWAS summary datasets to determine if there is a causal effect of higher genetically inherited RHR on the risk of AD. Additionally, using AD family history as a proxy for the diagnosis of AD, we analyzed the effect of genetically inherited higher RHR on the risk of having maternal or paternal family history of AD. We demonstrated that there was no evidence that supports a causal role of higher genetically predicted RHR in the risk of dementia.

To further substantiate our datasets, an LD Score regression (LDSC) analysis was conducted, revealing a genetic correlation of 0.13 between RHR and AD (standard error = 0.077, P = 0.087, Supplemental Materials). This correlation, however, was not supported by MR results as indicative of a causal relationship. Consequently, we posit that the association between RHR and AD risk may primarily arise from pleiotropy, aligning with conclusions in other cited literature. Our findings, robust to adjustments for RHR modifying medication, suggest no causal role for higher genetically predicted RHR in dementia risk.

There may be several explanations for our findings. Previous studies have demonstrated that higher RHR could more likely result in asymptomatic (paroxysmal) episodes of atrial fibrillation, leading to subclinical brain damage which is strongly associated with dementia (Haring et al., 2020). Alternatively, a higher RHR may occur secondary to CVDs with activation of sympathetic nervous system, where the RHR is a representative marker of performance of the excitability of sympathetic nervous system. Indeed, one previous study on the genetic link between the RHR gene SNP and dementia showed that the T allele of rs17070145 was associated with lower cognitive function, but was not associated with RHR (Wersching et al., 2011). Another study indicated that three SNPs (rs12474609, rs10201482 and rs980286) in the low density lipoprotein receptor-related gene may have a cognition-protective causal effect on AD, although they are not related to genetic susceptibility of RHR (Poduslo, Huang & Spiro, 2010). These results are consistent with our studies showing no evidence of causal effect of RHR on the risk of dementia, and their associations observed in observational studies were probably due to other underlying subclinical disease processes.

Dementia is a broad term used to describe a set of symptoms that impact memory, cognition, and social abilities. AD is one of the most common causes of dementia and may contribute to 60–70% of cases (Iturria-Medina et al., 2016). Although mounting evidence suggests a causal relationship between of CVDs and the risk of cognitive decline, the association between RHR and risk of dementia remains poorly understood. Previous studies have linked heart rate variability (HRV), calculated from RHR signals, with cognitive decline and dementia. The heart rate variability (HRV) is a marker of cardiovascular autonomic control (Thayer et al., 2009). Decreased HRV indexes indicate a low vagal activity that is associated with the development of several diseases such as diabetes, cardiovascular disease and cancers (Thayer, Yamamoto & Brosschot, 2010). Nevertheless, this does not necessarily imply that the results can be replicated in exploring the association between RHR and dementia. Recent analysis of the modified HRV, among a population of almost 1,500 participants, has revealed that the prognostic power RHR and HAV seems to be different for different outcomes (Sacha et al., 2013). With the HRV independent on RHR, the modified HRV rather than RHR proved to be independent risk factors of the dementia (Sacha et al., 2013).

Several existing studies have investigated the association of RHR with dementia and cognitive function (Bohm et al., 2012; Wang et al., 2019; Haring et al., 2020; Singleton et al., 2021). For example, the SNAC-K study (n = 2,147, average age 71 years) suggested that a higher RHR (≥80 bpm) was associated with the risk of dementia (hazard ratio (HR) 1.55). The association is still significant after participants with prevalent and incident CVDs were excluded. Similarly, RHR ≥ 80 bpm was also associated with cognitive decline (Mini-Mental State Examination score, β = −0.13). The ARIC study on middle-aged adults (average age 58 years) without prevalent CVDs showed that RHR ≥ 80 bpm (vs. <60 bpm) was associated with an increased risk of incident dementia and RHR ≥ 70 bpm was associated with a subsequent cognitive decline after the adjustment for various CVDs (β = −0.12), which was consistent with the SNAC-K study. The SPRINT study (n = 5,168, average age 67 years, mean RHR 70.4 beats per minute) indicated that a higher RHR (per ten beat-per-minute increase) was correlated with an increased risk of dementia or mild cognitive impairment (HR 1.09) in adjusted models. Finally, PRoFESS trial on 20,165 patients with post-ischemic strokes showed that an elevated RHR was associated with cognitive decline during the follow-up. Despite of the adjustment of CVDs in all analyses, none of these studies adjusted for the use of RHR medications, e.g., use of beta-blockers and calcium-channel blockers, which could affect the robustness of the results.

Conversely, the results were inconsistent among other two studies (Kuźma et al., 2018; Kokkinidis et al., 2020). The association of higher RHR with cognitive impairment after a 15-year follow-up was not observed from the Women’s Health Initiative Memory Study of participants without prevalent CVDs and cardiovascular risk factors (n = 493, age ≥ 63 years, average age 69 years), although there was an association between higher RHR and ischemic brain lesions on magnetic resonance imaging. RCT also showed that intervention using ivabradine (a specific inhibitor of RHR) to reduce RHR could not improve participants’ cognitive function. These contradictory results could be derived from the facts that RCT studies did not reach statistical power, and observational studies analyze correlation, but not causation.

A major concern we considered when planning this study was the potential confounding from the use of RHR inhibitors. Medication with RHR inhibitor has a modifying effect on the exposure, which may lead to the reverse causal effect between RHR and the risk of AD risk (Eppinga et al., 2016). Our MR results remained unaffected after the adjustment by excluding participants who were taking RHR medications. To the best of our knowledge, this is the first study that explored the causal effect of higher RHR on the risk of dementia. However, this study has several limitations. First, we used self-reported family history of dementia from UKBB participants as a proxy for AD diagnosis. The limitations of using the UKBB include self-reporting bias and use of a proxy phenotype rather than clinical diagnosis of AD, although there is a significant genetic correlation between proxy AD and clinical AD status. Nevertheless, a global meta-analysis demonstrates that self-reported information of family history of AD can indeed accurately reflect proxy for clinical AD diagnosis (Marioni et al., 2018). Second, our analyses did not reach sufficient statistical power to detect a causal effect, and thus further analysis using larger sample sets is needed to provide more conclusive results. Lastly, since there is a lack of enough information to estimate the actual overlap between exposure and outcome, weak instrument bias cannot be ignored. However, the additional R2 we provided may partially address this concern.

Conclusions

The present study did not observe evidence that supports causal effect of RHR on the risk of dementia, and the results remained stable after adjusting for RHR-modifying medication use. Association between increased RHR and dementia found in previous studies could be due to the underlying subclinical CVDs processes and cardiovascular risk factors. Further study using updated data from large genetic studies is warranted to verify the results of our MR study. Long-term randomized controlled studies are needed to determine if RHR medications could prevent dementia and be recommended as preventive agents.

Supplemental Information

Supplemental Information 1 Supplementary file for publication.

Supplemental Information 2 Raw dataset.

This work was made based on the generous sharing of GWAS summary statistics. We thank the participants, researchers, and staff associated with the many other studies from which we used data for this report. We thank the UK Biobank for providing summary statistics publicly for all researchers. We also thank the IGAP for providing summary results data for generating our results. The investigators within IGAP contributed to the design and implementation of IGAP and/or provided data but did not participate in analysis or writing of this report. IGAP was made possible by the generous participation of the control subjects, the patients, and their families. IGAP have made an excellent work on Alzheimer’s Disease and other neurodegenerative diseases.

Additional Information and Declarations

Competing Interests

Author Contributions

Data Availability

Kenta Nakai is an Academic Editor for PeerJ.

Xingxing Chen analyzed the data, prepared figures and/or tables, authored or reviewed drafts of the article, and approved the final draft.

Yi Zheng performed the experiments, analyzed the data, prepared figures and/or tables, and approved the final draft.

Jun Wang performed the experiments, prepared figures and/or tables, and approved the final draft.

Blake Yue analyzed the data, authored or reviewed drafts of the article, and approved the final draft.

Xian Zhang performed the experiments, prepared figures and/or tables, and approved the final draft.

Kenta Nakai conceived and designed the experiments, authored or reviewed drafts of the article, and approved the final draft.

Lijing L Yan conceived and designed the experiments, authored or reviewed drafts of the article, and approved the final draft.

The following information was supplied regarding data availability:

The raw data and the code are available in the Supplemental Files.

The datasets used for this article are available at:

- GWAS on family history of Alzheimer’s disease in UK Biobank: https://datashare.is.ed.ac.uk/handle/10283/3364.

- IGAP RV Summary Stats-Kunkle P-Value Data: https://www.niagads.org/igap-rv-summary-stats-kunkle-p-value-data.

- IGAP AD GWAS dataset: https://www.niagads.org/datasets/ng00036, DOI: 10.1038/ng.2802.

- Resting heart rate GWAS dataset: https://data.mendeley.com/datasets/czf5khpyhh/1.

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
