# Peer review of "Resting heart rate and risk of dementia: a Mendelian randomization study in the international genomics of Alzheimer’s Project and UK Biobank"

_PeerJ, doi:10.7717/peerj.17073_

## Round 0.1 · original submission · Major Revisions

Both of the reviewers raised major concerns. The authors should carefully address them.

·

Basic reporting

The article is written in English and clearly expresses a sufficient introduction and background and shows that the causal relationship between RHR and AD is really important. The structure of the manuscript is in line with the requirements of the journal. But I think some small but important issues need to be addressed, including 2 aspects.
(1) This article may need to improve the expressions due to unclear/ambiguous/wrong expressions. Some examples are the following:
1) Line 40. “Observational studies have demonstrated that resting heart rate (RHR) is associated with an increased risk of dementia.” RHR is the number of times our heart beats per minute when in a resting state, everyone has an RHR, it is an item of body measurement, not a risk factor for anything. I thought the writer wanted to express that “Higher RHR is associated with an increased risk of dementia” after I had read through the manuscript. If so, I suggest rewrite the sentence for easier understanding.
2) Lines 44-46. “…genetically predicted higher RHR…”. I have searched “genetically predicted higher RHR” in Google/Google Scholar/Pubmed but found nothing about it. However, I found an expression of “higher genetically predicted RHR” (Huang2022, DOI: https://doi.org/10.1210/clinem/dgab847) and “genetically predicted RHR” (Ma2023, DOI: 10.3389/fcvm.2023.1110231. Eppinga2016, DOI: https://doi.org/10.1038/ng.3708) from Pubmed. So it seems better to use “ higher genetically predicted RHR” than “genetically predicted higher RHR”.
3) Lines 46-48. “…effects of RHR against following different outcomes….” It is better to use “effect of X on Y” to replace “effect of X against Y”.
4) Lines 48-50. “…meta-analysis including these three GWAS results.” There only 2 not 3 datasets be listed before, does it a slip of the pen?
5) Lines 52-53. “The results of GSMR showed no significantly causal associations of genetically predicted higher RHR and the risk of AD (βGSMR = 0.12, P = 0.30).” Nouns should not be modified by adverbs, they should be adjectives. using “significant” instead of “significantly” here. Another issue is that the writer just lists one association statistic in the sentence (βGSMR = 0.12, P = 0.30), using “association” instead of “associations”. The last issue is the use of the prepositional phrase “…associations of X and Y…”, it seems not to be the right way. A similar situation appears in lines 101-102 “… causal association of X on Y…”. In general, the prepositional phrase including “association” could be “association of X with Y” and “association between X and Y”. Meanwhile, in the other part of the manuscript, “association between X and Y” or “relationship between X and Y” in lines 44-46, 95-96, 105-106, 143-145, 239-240, 260-264, 311-313 and 376-378 not clear enough to tell readers what the current study focus. Since the current study only investigates the causal effect of RHR on AD, I suggest using “effect of X on Y” to replace the corresponding phrases, and of course, the corresponding sentences need a little adjustment.
6) Lines 57-58. From the results of the current study, we can say there is no evidence supporting a causal relationship between RHR and dementia, neither a causal effect of RHR on dementia, nor a causal effect of dementia on RHR. However, it is still unclear whether dementia has a causal effect on RHR because the current study did not test the causal effect of dementia on the RHR. So it is better to express “Our study did not find evidence that supports a causal effect of RHR of dementia”, which is clearer and more accurate.
7) From the “2.3. Mendelian Randomization Analysis” section, we know that the authors used the GSMR model to conduct primary causal inference and used two kinds of IVWs and the MR-Egger regression method to conduct complement inference. But Figure S2 of the Supplemental Materials shows the scatter plot of MR analysis with methods of IVW (only one kind of the IVWs method introduced in the manuscript), MR-egger regression, simple mode, weighted median, and weighted mode. Why the information is inconsistent?
8) Lines 266-268. Readers are not authors, when the authors want to show a forest plot to readers, the authors need to tell clearly which figure readers need to look at. So please tell the readers which figure to look at directly.
9) There is a slip of the pen about “Cochran’s Q test” in lines 162, 186, 242, 253, and 264.

(2) This article may need to support more background information about inferences. Some examples are the following:
1) Lines 58-59. From the results of the current study, there is no evidence to support the causal effect of RHR on dementia. Where is the evidence to conclude that the previous observational association between RHR and dementia is due to the association between RHR and other cardiovascular diseases?
2) Lines 133-135. Here, the author writes that a genetic association study could determine the accuracy and reliability of self-reported measurement of family history of AD. How? Maybe need more information.

Experimental design

The current study is an original primary research within Aims and Scope of the journal, and with research question well defined, relevant, and meaningful. It is commendable that the authors used GWAS summary statistics based on big data to estimate the causal effect of RHR on AD, with several robust MR methods, and also conducted necessary sensitivity analysis to test the reliability of the results. However, the manuscript may need more detailed information about methods to address some important points, not excluding the fact that there are some issues that only I don't understand. Some examples are the following:
(1) Lines 119-138. First, let us name RHR GWAS summary statistics as Data-A, GWAS summary statistics of IGAP-AD as Data-B, and GWAS summary statistics of family history of AD of UKBB as Data-C. According to the introduction and URL for information on Data-A, I found both Data-A and Data-C based on UKBB. So there are some questions about discovery-replication design and MR analysis. An essential point in discovery-replication design is that the datasets of discovery and replication should be independent, i.e., without 0 sample overlap. Is it reasonable to use Data-A and Data-B to perform a discovery analysis, and then use a meta data combined of two data sets (i.e., Data-A and Data-C) and Data-B to perform a replication analysis? For MR analysis, the exposure dataset and outcome dataset should be independent as well. At least not overlap too much so that some tools still could perform an MR analysis. So, did the authors of the current study ever notice that and estimate the sample overlap between Data-A and Data-C? If so, how much sample overlap between them? and how the authors control it.
(2) Lines 142. “…all selected SNPs were previously confirmed….” Confirmed what here? The validity of SNPs as instrumental variables or something else?
(3) Lines 143-148. “…According to the assumptions of MR analysis…to examine…observed associations between genetic instruments and AD and remove potential confounders, …SNPs that have been reported to be associated with dementia…”. Does this sentence say that removing the SNPs associated with AD could not only remove the SNPs associated with AD but also the SNPs associated with confounders? From the supplementary material, we know that the author removed SNPs associated with one confounder— high blood pressure additionally. Another issue that needs to be considered is whether does makes sense to control the heterogenety and pleiotropy of SNPs in this way. The three assumptions of MR analysis are assumptions, and there is no way to test the second and third assumptions directly, but we can test the consequences of the assumptions by sensitivity analysis. Burgess and his colleague have discussed the selection of instrumental variables here (Burgess 2019, Doi:10.12688/wellcomeopenres.15555.3).
(4) Lines 152-153. “We scaled MR estimates per 10 bpm variation of the modifiable risk factors in all analyses.” How to perform the scale? And why to perform it?
(5) Lines 158-159. “…two-sample MR analysis was performed using beta value of the published RHR GWAS.” Just using the beta value is not enough to perform a Two-sample MR in the GSMR tool, please check the analysis details.
(6) Lines 159-160. “(Supplemental materials) as exposure for RHR and the following beta values and standard errors as outcome for AD.” Is it a complete sentence?
(7) Lines 161-162. “…Snps that show pleiotropy by the HEIDI-outlier instruments analysis and heterogeneity by the modified Cochran Q test were excluded (P < 0.01).” Why the threshold of P value is 0.01, not 0.05?
(8) Lines 164-166. To examine the robustness of the results and the possibility of reverse causal association, why not perform a reverse MR (i.e., AD as exposure and RHR as outcome)? The performance adopted in the current study “we also conducted analyses by excluding individuals who were using RHR modifying medication” seems not feasible. The issue that could be addressed is whether the medication affects the causal effect of RHR on AD, but not whether AD has a causal effect on RHR.
(9) Lines 167-168. The author uses the 1000 Genomes Project reference dataset to get an LD pattern but uses dbSNP151 for harmonizing data (Lines 148-149). Why not use the 1000 Genomes Project reference dataset to harmonize data also?
(10) Lines 168-169. What is the clumping window size and what kind of tool is used by clump in the current study?
(11) Lines 176-180. The horizontal pleiotropy in MR analysis could be tested by performing MR-Egger regression and considering whether the MR-Egger regression interception was different from 0. However, there is none of the comparisons of the results between the IVW and MR-Egger regression business. Why do the authors think making a comparison of results between the IVW and MR-Egger regression could determine if the MR-Egger interception was different from zero?
(12) Lines 190-192. In the current study, the authors used the GSMR model to conduct primary causal inference and used other methods including IVW as complements. When calculating MR power, the authors calculate for a complement method (i.e., IVW method), but not the primary method (i.e., GSMR model).
(13) Lines 195-196. The authors may get the wrong understanding from the listed reference. In the listed reference (i.e., Burgess,2011), Burgess and his colleague did not agree that “F statistics >10 could avoid the weak instrument bias in MR studies”, on the contrary, they concluded that “Data-driven choice of instruments or analysis can exacerbate bias. In particular, any guideline such as F > 10 is misleading.”
(16) Lines 258-259. How to perform a meta-analysis with a combination of IFAP and UKBB family history data sets?

Validity of the findings

The research question is important for us to understand the causal effect of RHR on AD. However, I doubt the validity of the findings, not only because of the issues listed above but also because of some problems listed below:
(1) Validity of the primary causal inference. The results showed that the primary and complementary causal inferences are not significant, but in the primary causal inference, the causal effect of RHR on AD risk is positive, while in the complementary causal inferences, the causal effect of RHR on AD risk is negative. Why do they show a conflict effect? Is it possible something wrong in the analysis process such as a wrongly defined effect allele in either primary inference or complement inference?
(2) Lines 272-273. What is the standard used in the current study so that could think 28.3% is moderate power?
(3) Lines 275-278. F-statistic is used to determine how strong the instrument variable is, each instrument variable could get one F value and use 10 as a reference thresholding of weak or strong, but also could sum them up to get an overall value but not compare with 10 anymore. In Table 2, the author gives an overall value, it could not be compared with 10 to determine whether the study has sufficient power to detect a causal effect. On the contrary, the power of analysis to detect a causal effect could be calculated described in the method section. As shown in Table 2, the power to detect a causal effect of RHR on AD is 28.3%, which is low. I think 28.3% is far from sufficient power, and suggest the author reconsider it. Additionally, the authors used 5 × 10-8 to choose instrumental variables from GWAS summary statistics of exposure, the corresponding F statist of each instrumental variable is about 30, and there is no need to check whether the F value of each instrumental variable is bigger than 10.
(4) Conclusion in Abstract: From the results of the current study, we can say there is no evidence supporting a causal relationship between RHR and dementia, neither the causal effect of RHR on dementia, nor the causal effect of dementia on RHR. However, it is still unclear whether dementia has a causal effect on RHR. The current study did not test the causal effect of dementia on the RHR. So in Lines 57-58, it is better to say “Our study did not find evidence that supports a causal effect of RHR of dementia”, which is clearer and more accurate.
(5) Lines 415-416. The URL (http://www.cardiomics.net/download-data) of the resting heart rate GWAS dataset does not exist.

Additional comments

Thanks to the authors for researching the causal effect of RHR on AD. It is a nice work with a nice idea but needs some improvements. I don’t want to hurt the author, but I think the analysis process is not convincing enough to conclude whether the RHR has a causal effect on AD. I hope I have listed all the points that could help to improve the authors’ work and manuscript, and hope that the authors will improve their work and refresh our knowledge about the causal effect of RHR on AD with more convincing evidence.

Reviewer 2 ·

Basic reporting

1. The exposure chosen for the article is a common phenotype, not a disease, and it is innovative to use Mendelian randomization to study its causal relationship with exposure in this way.
2. This paper uses two-sample Mendelian randomization to explore the relationship between resting state heart rate and dementia, however, the paper also mentions that resting state heart rate may cause dementia due to the same pathophysiological changes as cardiovascular disease, but does not shed much light on the unique advantages of resting state heart rate in the introduction and discussion sections

Experimental design

1. for the GWAS data used for exposure, the article says it is based on 64 SNPs, may I ask if this is extracted from previous studies or the original data, if it is extracted from previous studies, does it therefore miss important instrumental variables in the original data, and are there any previous MR studies that also used 64 SNPs in the same exposure scenario
2. Further test whether the SNPs are pleiotropic. It suggests that MRPRESSO analysis should be provided.
3. This paper needs to emphasize the sample overlap and race situation
4. Since the results of this paper are negative, it is recommended to do bidirectional Mendelian randomization to determine the robustness of the results
5. Among the statistical methods in the article, only two methods, MR-Egg and IVM, are used in this article, and more methods should be used

Validity of the findings

1. In this paper, negative results are attributed to possible cardiovascular factors, and it is therefore recommended that relevant risk factors such as obesity, smoking, and other exposures be included in order to perform multivariate Mendelian randomization to exclude the effect of their SNP pleiotropy.
2. In the Discussion section, line 338, "Decreased HRV indexes indicate a low vagal activity that is associated with the development of several diseases such as Decreased HRV indexes indicate a low vagal activity that is associated with the development of several diseases such as diabetes, cardiovascular disease, cancers, and Alzheimer's disease (AD). Nevertheless, this does not necessarily imply that the results can be replicated in exploring the association between RHR and dementia." This sentence is confusing because most GWAS used in this paper were about AD disease. So the relationship between AD and dementia needs to be clearly elucidated.
3. LDSC or other related methods should be added that can show a genetic correlation between the exposure and outcome

---

## Round 0.2 · Minor Revisions

There are still some minor comments.

·

Basic reporting

The article has been greatly improved and is generally in line with the requirements of the journal.

Experimental design

It would be a better to complement the reliability of the primary causal inference with non-overlap between exposure and outcome dataset, but it is a pity that such data is unaviable for the authors currently.

Validity of the findings

The research is helpful for readers to understand the causal effect of RHR on AD, but I think two small but important issues still need to be addressed.

(1) There are several tools to calculate power for a specific MR method. A widely used one was listed in your response “(https://sb452.shinyapps.io/power/) (Burgess, 2014)”. In fact, in the supplementary material of “(Burgess, 2014)”, the authors have also supplied another way to calculate power in the R platform, but just be careful the “b1” in the formula is absolute value form. The contradictory point is that the author says “it appears that the existing tools do not support power calculations specific to certain MR methods such as GSMR or IVW”, but lists the website you used to calculate power for the estimate by IVW. As I know, GSMR returns the beta of the estimate, corresponding standard error, and SNP index, so that by combining the information about your dataset, you can easily calculate power for each MR method. I still recommend you supply the power of GSMR because it is the method that you used to perform primary causal inference.

(2) Previous research has found that in two-sample MR analysis, the bias of estimate is associated with sample overlap, especially with weak IVs (Burgess 2016, doi: 10.1002/gepi.21998). For the current study, since there is a lack of enough information for the authors to estimate the actual overlap between exposure and outcome in sensitivity analysis, I recommend that the author supply the R2 (i.e., a measure of how much variance of exposure can be explained by the IVs used to performed MR analyses), and list sample overlap issue in limitation.

Additional comments

Thanks to the authors for researching the causal effect of RHR on AD. It is a nice work with a nice idea,I hope I have listed all the points that could help to improve the authors’ work and manuscript.

Reviewer 2 ·

Basic reporting

The article has been answered and improved as required

Experimental design

The article has been answered and improved as required

Validity of the findings

The article has been answered and improved as required

Additional comments

The article has been answered and improved as required

---

## Round 0.3 · Minor Revisions

The authors should double check their results.

·

Basic reporting

The article has been answered and improved as required.

Experimental design

The article has been answered and improved as required.

Validity of the findings

There is concern regarding the power of the GSMR. The two methods mentioned for calculating power in an MR analysis result estimated by the same method will yield the same power. The distinction between these two methods is that the website used OR, while the R code used the absolute value of β. For instance, when estimating the causal effect of genetically predicted RHR on AD, the β value is 0.12 (as shown in lines 218-220 of the manuscript), and the corresponding OR is approximately 1.127497 (R code, exp(0.12)). Table 2 indicates that the proportion of cases in the outcome dataset is 0.31, and the corresponding ratio of cases to controls is about 0.31/0.69 = 0.4492754. Furthermore, the proportion of exposure explained by IVs is 0.20%, and the outcome sample size is 54,162. Using these parameters in the website (https://sb452.shinyapps.io/power/) to calculate power results in a power of 8.3% with a significance level of 0.05 (the snapshot of the calculation will be submitted with a pdf file). Similarly, the power can be calculated in R code as shown below, resulting in a power of 0.08343443, which can be rounded to 8.3%. It is unclear how the authors calculated the power and obtained a significantly different result from 8.3%.

# ************ R code to calcualte the power of the instance
samplesize = 54162
rat_cas_con = 0.31/0.69
b = 0.12
sig =0.05
R2 = 0.002
pnorm(sqrt(samplesize*R2*(rat_cas_con/(1+rat_cas_con))*(1/(1+rat_cas_con)))*abs(b)-qnorm(1-sig/2))
# *************

Additional comments

The article has been answered and improved as required.

---

## Round 0.4 · accepted · Accept

All the concerns have been addressed. The manuscript can be accepted now!